# ADAPT: Multimodal Learning for Detecting Physiological Changes under Missing Modalities

**Julie Mordacq**[1,2,3]                                        JULIE.MORDACQ@INRIA.FR
[1] *Inria Saclay, France*
[2] *Ecole Polytechnique, France*
[3] *LIX, CNRS, IP Paris, France*
**Leo Milecki**[1,4]                                        LEO.MILECKI@CENTRALESUPELEC.FR
[4] *MICS, CentraleSupelec, Paris-Saclay University, France*
**Maria Vakalopoulou**[1,4]                        MARIA.VAKALOPOULOU@CENTRALESUPELEC.FR
**Steve Oudot**[*1,2]                                        STEVE.OUDOT@INRIA.FR
**Vicky Kalogeiton**[*2,3]                                VICKY.KALOGEITON@POLYTECHNIQUE.EDU

**Editors:** Accepted for publication at MIDL 2024

## Abstract

Multimodality has recently gained attention in the medical domain, where imaging or video modalities may be integrated with biomedical signals or health records. Yet, two challenges remain: balancing the contributions of modalities, especially in cases with a limited amount of data available, and tackling missing modalities. To address both issues, in this paper, we introduce the **A**nchore**D** multimod**A**l **P**hysiological **T**ransformer (ADAPT), a multimodal, scalable framework with two key components: (i) aligning all modalities in the space of the strongest, richest modality (called *anchor*) to learn a joint embedding space, and (ii) a Masked Multimodal Transformer, leveraging both inter- and intra-modality correlations while handling missing modalities. We focus on detecting physiological changes in two real-life scenarios: stress in individuals induced by specific triggers and fighter pilots' loss of consciousness induced by $g$-forces. We validate the generalizability of ADAPT through extensive experiments on two datasets for these tasks, where we set the new state of the art while demonstrating its robustness across various modality scenarios and its high potential for real-life applications. Our code is available at https://github.com/jumdc/ADAPT.git.
**Keywords:** Multimodality, Missing Modalities, Contrastive Learning, Biomedical signals

## 1. Introduction

Monitoring physiological changes to external stimuli is crucial for assessing individuals' well-being, particularly in contexts with medical and safety implications. Examples include stress, a response to emotional, mental, and physical challenges (Schneiderman et al., 2005), and a triggering or aggravating factor for various pathological conditions (Dimsdale, 2008). High-performance environments, such as exposure to $g$-forces in aircraft, can lead to alterations in consciousness (Morrissette and McGowan, 2000). At the same time, drowsiness during driving poses a critical physiological response with safety implications, contributing to road accidents and fatalities (Stewart, 2023). Various sensors report physiological

---

*  Equal supervision

changes that may be detected visually (videos), acoustically (audio), or from biomedical signals (e.g., electrocardiograms). However, specific modalities may be missing during training and testing. Therefore, developing methods capable of handling missing modalities during both stages while balancing modalities' contributions is crucial to ensure robustness, notably when modalities with strong unimodal performances are severely missing.

Various methods address the challenge of missing modalities, each with notable limitations, including (1) bias towards the most available modalities leading to sub-optimal performance (Konwer et al., 2023), (2) dependence on complete modalities during training (Mallya and Hamarneh, 2022; Chen et al., 2021a) , (3) limited generalizability to more than two modalities (Ma et al., 2021, 2022), and (4) utilization of a shared encoder tailored for modalities with inputs of the same dimensions which complicates extension to heterogeneous modalities like imaging and biomedical signals (Konwer et al., 2023).

To address the above issues, we introduce the **A**nchore**D** multimod**A**l **P**hysiological **T**ransformer (ADAPT) that is designed to operate effectively under missing modalities both during training and inference enabling robust real-life applicability. ADAPT consists of two key components. First, our goal is to embed all modalities in the same feature space. Instead of optimizing one loss per modality pair, which would result in quadratic growth of training time, we align each modality to one frozen modality, called *anchor*. It allows learning a joint embedding space with linear scalability and balancing each modality's contribution. We call this step the 'anchoring'. Second, it comprises a Masked Multimodal Transformer that leverages inter- and intra-modality correlations to concatenate features from different modalities into a unified representation. Additionally, we leverage masked attention from the transformers (Vaswani et al., 2017) to ensure flexibility in handling missing modalities similarly to Ma et al. (2022); Milecki et al. (2022). When a modality is unavailable, its corresponding feature representation is masked. The transformer is trained using two objectives: self-supervised learning and the objective of the downstream task.

ADAPT is applied to the challenging task of detecting physiological changes using multimodal medical data with missing modalities during training and inference. Specifically, we focus on detecting alterations in pilots' consciousness induced by $g$-forces in fighter jets and stress triggered in individuals by specific stimuli (Chaptoukaev et al., 2023). We show that ADAPT outperforms the previous state of the art on both tasks and datasets while handling missing modalities. Extensive experiments demonstrate its robustness against missing modalities across various scenarios, highlighting its effectiveness for real-life applications.

Our contributions are: (i) ADAPT, a modular framework that aligns multimodal representations to a common rich feature space; (ii) a modality-fusion strategy to handle missing modalities both at training and inference time; (iii) we set the new state of the art on two tasks and datasets and provide extensive evaluations highlighting ADAPT's superiority.

## 2. Related Work

**Handling missing modalities.** Missing modalities pose a persistent challenge in Multimodal Learning, particularly in medical imaging, due to privacy concerns or impractical data acquisition (Liang et al., 2021; Azad et al., 2022). Various strategies have been explored to address this challenge. *Knowledge Distillation* (Hu et al., 2020; Mallya and Hamarneh, 2022; Wang et al., 2023b) involves learning from a teacher network trained on complete

modality data. *Generative modeling* aims to impute missing inputs by generating synthetic data (Yoon et al., 2018; Sharma and Hamarneh, 2019). Both approaches rely on complete modality at training, which can be insufficient for robust training. Another line of work is *common space modeling*, which learns a shared latent space from partially available modalities (Ma et al., 2021; Konwer et al., 2023; Wang et al., 2023a). SMIL (Ma et al., 2021) perturbs the latent feature space to approximate the embedding of missing modalities but is limited to bi-modal datasets, limiting its generalizability. ShaSpec (Wang et al., 2023a) addresses more than two modalities by learning shared and specific features, but its use of a shared encoder complicates generalization to heterogeneous modalities of different dimensions. Additionally, shared latent space modeling may introduce biases toward the most available modalities (Konwer et al., 2023). Simultaneously, cross-modal contrastive learning has shown impressive results (Zhang et al., 2022; Milecki et al., 2023; Li et al., 2023) by aligning multimodal data in a joint embedding space. Recently, ImageBind (Girdhar et al., 2023) aligned six modalities by relying on image-paired data and emphasized that aligning all pair combinations is unnecessary to bind more than two modalities together. Our proposed ADAPT advances this by training unimodal encoders solely with supervision from one modality, aligning them in a joint embedding space. It ensures that every modality contributes to the final representation, even if it is severely missing during training.

**Multimodal transformer.** Transformers (Vaswani et al., 2017) have become the de facto approach for multimodal tasks (Recasens et al., 2023; Srivastava and Sharma, 2024). They rely on the attention mechanism to model long-range dependencies with the flexibility to account for incomplete samples. Milecki et al. (2022); Ma et al. (2022) efficiently handle missing data in sequences and bimodal datasets through masked attention. ADAPT extends this to more than two modalities by leveraging attention to fuse them and exploring their inter- and intra-modal correlations while masking missing ones. We also perform a systematic study of missing modalities during training and testing, showing the versatility and potential of ADAPT for real-life scenarios.

## 3. Method

This study addresses the detection of physiological changes using multimodal data, including video, audio, and biomedical signals. Real-world scenarios often involve missing modalities, motivating our goal to develop a modality-agnostic representation with broad applicability and to propose ADAPT – **A**nchore**D** multimod**A**l **P**hysiological **T**ransformer. An overview of ADAPT is presented in Figure 1. **Notations.** Let $\mathcal{D} = \{(x_m^i)_{m=1}^M, y^i\}_{i=1}^N$ denote our training dataset, with $M$ modalities and $N$ labeled observations and $x^i = (x_m^i)_{m=1}^M$ the $i$-th observation (i.e., a family of $m$ modality values) with $y^i \in \mathcal{Y} = \{0, .., J\}$ its corresponding label (i.e., a physiological state). Given this input, we seek to train a neural network $\mathcal{F}$, that associates to any observation, with any missing modality, a target label $y \in \mathcal{Y}$.

### 3.1. Anchoring modality-specific encoders

We train modality-specific encoders with a contrastive learning objective to align their representations to the one of the *anchor*. In this work, anchor is the video, as it can capture visually distinguishable physiological changes; however, any modality can be the anchor.

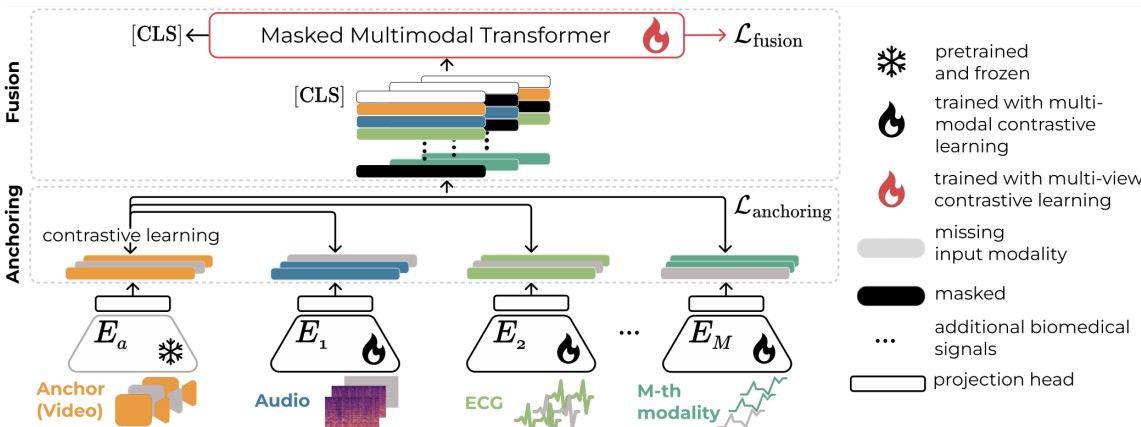

Figure 1: **Overview of ADAPT.** In each minibatch, ADAPT takes up to $M$ modalities, including video, audio, and biosignals, as input to produce a modality-agnostic representation for downstream tasks. It is trained in two steps. (i) *Anchoring.* We align the representations of all modalities via contrastive learning to the one of an *anchor* modality, i.e., the strongest and richest modality; here the video. (ii) *Fusion.* The encoders' features are concatenated and fed into the Masked Multimodal Transformer. When a modality is unavailable, the transformer masks its corresponding feature representations. The final representation (i.e., [CLS] token output) is used for downstream tasks.

**Modality-specific encoders.** Each modality is encoded using a dedicated encoder. For *video*, we use the pre-trained Hiera (Ryali et al., 2023) encoder. For *audio*, each sample is encoded into a mel-spectrogram (a 2D acoustic time-frequency representation of sound), fed to BYOL-A (Niizumi et al., 2021) to obtain a 1-d feature. *Biomedical signals* are processed using 1D CNNs (Wang et al., 2023c; Ismail Fawaz et al., 2019). We add a modality-specific linear projection head to each encoder to obtain a fixed size $d$ dimensional embedding. ADAPT can be extended to other modalities by adding their respective encoders.

**Anchoring.** We consider a pair of modalities with aligned observations $(\mathcal{A}, \mathcal{M}_m)$, where $\mathcal{A}$ represents the anchor (video) and $\mathcal{M}_m$ another modality. The anchor video $x_a^i$ and its corresponding observation $x_m^i$ are encoded using $z_a^i = E_a(x_a^i)$ and $z_m^i = E_m(x_m^i)$, respectively, where $E_a$ is a pre-trained and frozen video encoder and $E_m$ a DNN. Projection heads map the embeddings to $f_a^i, f_m^i \in \mathbb{R}^d$. The loss is computed on $f_a^i$ and $f_m^i$ (Girdhar et al., 2023):

$$\mathcal{L}_{\mathcal{A}, \mathcal{M}_m} = -\sum_{i=1}^{B} \log \frac{\exp(\cos(f_a^i, f_m^i)/\tau)}{\sum_{k=1}^{B} \exp(\cos(f_a^i, f_m^k)/\tau)} \quad , \tag{1}$$

where $\tau$ is a temperature parameter $\tau \in \mathbb{R}^+$, $\cos(.,.)$ the cosine similarity, and $B$ the batch size. In practice, we use a symmetric loss: $\mathcal{L}_{\mathcal{A}, \mathcal{M}_m} + \mathcal{L}_{\mathcal{M}_m, \mathcal{A}}$. To alleviate the modality gap (Liang et al., 2022), we add Gaussian noise to the modality $m$ representation (Gu et al., 2023). We use a cosine schedule for the temperature parameter (Kukleva et al., 2023). Given $M$ modalities, we define the anchoring loss as $\mathcal{L}_{\text{anchoring}} = \sum_{m=1, \mathcal{M}_m \neq \mathcal{A}}^{M} (\mathcal{L}_{\mathcal{A}, \mathcal{M}_m} + \mathcal{L}_{\mathcal{M}_m, \mathcal{A}})$.

## 3.2. Masked Multimodal Transformer

To effectively build modality-agnostic representations, we use the transformer (Vaswani et al., 2017) with $N_L$ attention blocks. For each sample, we stack the modality-specific representations, $f_m^i \in \mathbb{R}^d, \forall m \in [1, M]$, into a single matrix and prepend a special token [CLS], yielding a matrix $F \in \mathbb{R}^{(M+1) \times d}$. Similarly to Liu et al. (2022); Nagrani et al. (2021), the query, key and value are derived from $F$ via: $Q = W^Q F$, $K = W^K F$ and $V = W^V F$ where $Q, K \in \mathbb{R}^{(M+1) \times d_k}$ and $V \in \mathbb{R}^{(M+1) \times d_v}$. Our modelization of inter-modal interactions differs from the usual cross-attention (Chen et al., 2021b; Jaegle et al., 2021), which asymmetrically combines two separate embedding sequences of the same dimension. Using stacked features $F$ allows generalization to any number of modalities, with linear scalability in the number of modalities instead of quadratic.

**Handling missing modalities.** Inspired by Milecki et al. (2022) for missing follow-up patient examinations, we apply our strategy to deal with missing modalities to the scaled dot-product, core of each multi-head self-attention sub-layer. We consider one sub-layer with one head ($h = 1$) for simplicity. We use a masking binary matrix $Z \in \mathbb{R}^{(M+1) \times (M+1)}$ that specifies which modalities are missing: $z_{ij} = 1$ if $i$ and $j$ are available, else $z_{ij} = 0$. The output $O \in \mathbb{R}^{(M+1) \times d_v}$ of the attention mechanism is, for $O_i$ each line of $O$:

$$O_i = \sum_j z_{ij} \frac{\exp(Q_i^T K_j / \sqrt{d_k})}{\sum_{\{j', z_{ij'}=1\}} \exp(Q_i^T K_{j'} / \sqrt{d_k})} V_j \quad . \tag{2}$$

When $h > 1$, queries, keys, and values are linearly projected $h$ times with different, learned linear projections, concatenated, and once again projected after the scaled-dot product.

**Modality dropout.** We train the Masked Multimodal Transformer with a multi-view contrastive objective (Chen et al., 2020). Drawing inspiration from Shi et al. (2022), we mitigate the model's over-reliance on a single modality while enhancing its robustness in the absence of modalities through an augmentation technique called *modality dropout*. We leverage the masking scheme at the attention level to randomly mask input modalities. Given a batch $\mathcal{Z}$, we create two simultaneous view $\mathcal{Z}'$ and $\mathcal{Z}''$. For each observation within $\mathcal{Z}'$, we hide up to $M - 1$ modalities following a uniform probability, $M$ the number of modalities. Additionally, motivated by Han et al. (2020), who recently showed the effect of additive noise on electrocardiograms, we add $\epsilon \sim \mathcal{N}(0, \sigma)$ on the biomedical signals to each view independently. We chose $\sigma$ based on the amplitude of the signal. We use the infoNCE loss to enforce the similarity between the two views (Chen et al., 2020) on the final representation output given by the [CLS] token. Since the two representations are already mapped to the same dimension, following Jing et al. (2022), we directly optimize the representations to enforce scalability and mitigate dimension collapse: $\mathcal{L}_{\text{fusion}} = -\sum_{i=1}^B \log \frac{\exp(\cos(\text{CLS}^i, \text{CLS}^{i'})/\tau)}{\sum_{k=1}^B \exp(\cos(\text{CLS}^i, \text{CLS}^{k'})/\tau)}$.

## 4. Experiments & Results

### 4.1. Experiment Setting

**Datasets.** StressID (Chaptoukaev et al., 2023) for stress identification contains physiological responses via electrocardiogram (ECG), electrodermal activity, respiration, audio,

and videos. We use the training, val, and test splits provided by Chaptoukaev et al. (2023).
LOC. We present the Loss Of Consciousness dataset, collected during aeromedical training of flight personnel by the French Ministry of Armed Forces (Appendix A). It includes videos and biomedical sensor data: ECG, quadriceps electromyograms, acoustic breathing, pedal pressure, and self-reported visual field. It comprises 1666 launches with 416 subjects, split into train, val, and test sets in a 6:2:2 ratio based on patient ID. We employ 5-fold cross-validation and report the average. Each launch is labeled for consciousness alteration. The dataset exhibits a high imbalance ratio of {1:50}. In real life, videos are impractical due to pilots' equipment (helmets & $O_2$ masks), despite being the primary modality used by doctors to monitor launches during aeromedical training (see Appendix A).

**Missing modalities.** StressID has 18% and 46% of missing video and audio recordings, respectively. For LOC, videos are absent in 90% of observations. We denote the entire training and testing sets as $X_{\text{train}}$, $X_{\text{test}}$ (considering samples with and without missing modalities); and $X^*_{\text{train}}$, $X^*_{\text{test}}$ for the train and test sets where all modalities are available.

**Metrics.** We use the balanced accuracy (ACC) and weighted F1 score (F1). For LOC, to ensure robustness to class imbalance (Huang et al., 2016; Luque et al., 2019), we also report the true positive rate (TPR) as it ensures not missing out on pilots fainting; and the true negative rate (TNR) for completeness. We report metrics in the format `mean(std)` in %.

**Implementation details.** The Anchoring and Masked Multimodal Transformer are trained on $X^*_{\text{train}}$ and $X_{\text{train}}$, respectively. A linear classifier is trained using the [CLS] token for the final task. We train for 70 epochs using AdamW optimizer, a starting learning rate of $1e^{-4}$, followed by a cosine schedule and a linear warm-up of 4 epochs. Given their size difference, we set the batch size to 256 for LOC and 128 for StressID. To tackle LOC class imbalance, we use the Balanced Cross Entropy loss (Huang et al., 2016) (more in Appendix B.1).

### 4.2. Results

**Comparison to the state of the art** (Table 1) in the presence of all modalities. We compare ADAPT against unimodal baselines for video, audio, and biomedical signals concatenated (rows 1, 2 & 3), 'feature fusion' and 'decision-level fusion' (rows 4 & 5)(Chaptoukaev et al., 2023), ShaSpec+ (Wang et al., 2023a) (row 6) (more in Appendices B.2 and C).

For StressID, we observe that ADAPT outperforms all methods from Chaptoukaev et al. (2023) by a notable margin; for instance, it outperforms 'decision-level fusion' by 4% in ACC and 6% in F1. Additionally, it remains highly competitive with ShaSpec+.

For LOC, using only video (row 1) results in the best performance; however, video is unavailable in real-life scenarios. Instead, ADAPT handles missing modalities by leveraging representations from all modalities during training. Surprisingly, 'feature fusion', 'decision-level fusion', and ShaSpec+ lead to unbalanced metrics, i.e., they result in a high TNR while significantly sacrificing TPR (respectively 29.5%, 20.4% and 7.3%), showing they predict most samples as negative. This reveals their unsuitability for real-life cases with highly imbalanced classes where both TPR and TNR matter. Note that in our target scenarios, TPR is more important, as it is critical to detect pilots losing consciousness. By contrast, ADAPT results in a TPR of 69.5% (+40% vs. fusion methods) while maintaining a balanced TNR of 65.3%. This is further shown in Figure 2, where ADAPT (blue crosses) strikes the

| | LOC | | | | StressID | |
|---|---|---|---|---|---|---|
| | **ACC** | **F1** | **TNR** | **TPR** | **ACC** | **F1** |
| Video | **87.1(1.2)** | 98.0 (0.2) | 98.0(0.4) | **75.6(2.5)** | 62(4)[*] | 67(3)[*] |
| Biomedical signals | 72.0(2.0) | 50.0(2.3) | 94.0(2.0) | 42.1(1.3) | 58(4)[*] | 66(5)[*] |
| Audio | 57.6(1.5) | 96.0(0.2) | 95.5(4) | 19.7(2.9) | 62(4)[*] | 67(4)[*] |
| Feature Fusion (Chaptoukaev et al., 2023) | 79.3(2.5) | 63.9(9.6) | 97.0(0.9) | 29.5(20) | 61(3)[*] | 66(4)[*] |
| Decision-level fusion (Chaptoukaev et al., 2023) | 60.2(2.2) | **99.0(0.0)** | **100.0(0.0)** | 20.5(4.5) | 65(5)[*] | 72(5)[*] |
| ShaSpec+ (Wang et al., 2023a) | 53.4(5.1) | 97.3(1.0) | 99.4(1.0) | 7.3(10.4) | **70.2(3.7)** | 75.7(5.3) |
| ADAPT | 67.4(1.2) | 76.9(2.5) | 65.3(1.6) | 69.5(2.0) | 69.5(3.7) | **75.9(4.3)** |

Table 1: **Comparison of ADAPT to SOTA on** $X^*_{\text{test}}$. Gray-out denotes the video modality, which is impossible to gather in real-life. **Bold**, underlined indicate the top **1**, 2 performing, respectively.
[*]Results from Chaptoukaev et al. (2023).

best balance between TPR and TNR vs. other methods (red crosses). This testifies to ADAPT not being misled by the high class imbalance.

**Robustness to missing modalities** (Table 2). We first report baseline results (row 1) on the default test set $X_{\text{test}}$, i.e., no modality removed in StressID and 90% of videos missing for LOC. Then, we completely remove one or two modalities from $X_{\text{test}}$ and compare the results ($\Delta$) to the ones obtained on $X_{\text{test}}$.

For LOC, ADAPT shows robustness in all scenarios, with a $|\Delta|<8\%$ and average $|\Delta|=2.6$ compared to the baseline. This is further shown in Figure 2, where the balance between TNR and TPR (blue circles) remains consistent across all scenarios. Interestingly, for *no-video*, even though video-only provides strong unimodal performance (Table 6), ADAPT maintains high performances, indicating its capability of aligning representations in the video (anchor) space. Furthermore, for the *real-life* scenario where we remove both video and visual field (row 2), the results remain competitive with an average $|\Delta|=2.72\%$, even though these modalities individually perform the best (Table 6). Additionally, *no-audio*

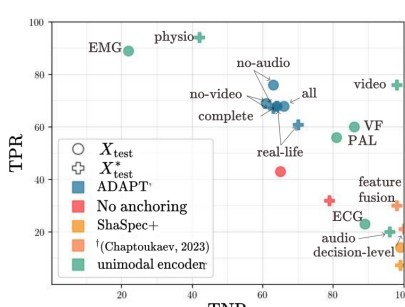

Figure 2: **TPR vs TNR for LOC.** [†]Methods from (Chaptoukaev et al., 2023)

demonstrates consistent results, keeping the TNR and TPR balanced (see Appendix C).

For StressID, we remove audio and/or video, the most cumbersome modalities to acquire and examine the *no-audio*, *no-video* and *real-life* (i.e., no audio, no video) scenarios. The variation remains consistent for both *no-audio* and *no-video*: $|\Delta|<8.3\%$. However, it is more consequent for *real-life*, with a significant drop in TPR for an equivalent TNR, as expected as we remove the richest modalities.

Overall, even by removing modalities, ADAPT successfully detects stress or loss of consciousness with more than 60% ACC and more than 50% TPR, highlighting its ability to handle missing modalities, in contrast to all other methods unable to address this.

**Ablations. 1. Impact of the anchoring before fusion and choice of anchor** (Table 3). Anchoring with video shows significant benefits, particularly in LOC with an 11.6% increase in ACC alongside consistent F1 scores. Similarly, for StressID, anchoring improves

| | LOC | | | StressID | | |
|---|---|---|---|---|---|---|
| | **ACC** $+\Delta$ | **TNR** $+\Delta$ | **TPR** $+\Delta$ | **ACC** $+\Delta$ | **TNR** $+\Delta$ | **TPR** $+\Delta$ |
| | 66.2(3.3) | 64.4(5.9) | 68.0(4.2) | 69.5(2.9) | 61.9(6.9) | 77.1(6.3) |
| *real-life* | 62.0(3.2) 4.2 | 60.8(1.0) 3.6 | 68.5(4.3) -0.5 | 60.0(4.8) 9.5 | 66.9(9.5) -5.0 | 53.1(7.9) 24 |
| *no video* | 67.1(2.2) -1.1 | 66.2(3.9) -1.8 | 68.0(2.0) 0.0 | 61.2(4.6) 8.3 | 53.7(10.1) 8.5 | 68.8(7.1) 8.3 |
| *no audio* | 69.5(5.4) -3.3 | 63.0(12.0) 1.4 | 76.1(4.9) -8.1 | 68.3(2.9) 1.2 | 65.8(6.8) -4.0 | 70.7(7.4) 6.4 |

Table 2: **Evaluation of ADAPT on three modality scenarios on $X_{\text{test}}$.** For each scenario, we remove one or two modalities from the test samples. We report mean and standard deviation for ACC, TNR, and TPR, and the differences ($\Delta$) compared to tests without removed modality (first row).

| Anchor | | LOC | | StressID | |
|---|---|---|---|---|---|
| **Audio** | **Video** | **ACC** | **F1** | **ACC** | **F1** |
| ✗ | ✗ | 54.6(1.0) | 78.4(4.1) | 65.8 (1.8) | 65.9 (1.7) |
| ✓ | ✗ | 54.0(2.1) | **78.5 (1.4)** | 63.6(4.8) | 63.3(5.0) |
| ✗ | ✓ | **66.2(3.3)** | 78.0(4.3) | **69.5(2.9)** | **69.6(3.1)** |

| Features | Anchoring | Fusion | ACC | F1 |
|---|---|---|---|---|
| [‡]handcrafted features | ✗ | [†]Feature level | 61(3)[*] | 66(4)[*] |
| | | [†]Decision level | 65(5)[*] | 72(5)[*] |
| | ✓ | ADAPT | 51.5(2.3) | 60.2(4.7) |
| ADAPT | ✓ | [†]Feature level | 65.2(7.2) | 71.2(10) |
| | | [†]Decision level | **70.7(3.3)** | **78.8(2.9)** |
| | | ADAPT | 69.5(3.7) | 75.9(4.3) |

Table 3: **Ablation study of *anchoring* on $X_{\text{test}}$.** We report the results with anchoring prior fusion (considering the audio or the video as the anchor) and without. **Bold** indicates the top **1** performing.

Table 4: **Study of ADAPT components with SOTA for StressID on $X^*_{\text{test}}$.** [‡]Handcrafted features, [†]Methods, [*]Results from Chaptoukaev et al. (2023). **Bold** and underlined indicates the top **1**, 2.

both ACC and F1 by 3.7%. Any anchor may be considered; we explore using the audio (row 3), but it leads to suboptimal performances. Overall, the anchor selection is driven by its robust unimodal performance, which remains effective despite high missing modalities.
**2. Impact of feature configurations and fusion methods** (Table 4). Compared to the 'feature fusion' and 'decision-level fusion' (rows 1,2, Chaptoukaev et al. (2023)), our features and fusion method (last row) significantly increase ACC and F1 by 5.7% and 6.8%, respectively, further highlighting the advantages of *anchoring*. We also investigate applying *anchoring* to features from Chaptoukaev et al. (2023) (row 3) by solely training the projection head, as opposed to both the encoder and projection head. Although this yields decent results, the inability to learn features optimally is a drawback. Finally, the ADAPT entire pipeline (row 6) delivers competitive results while accommodating missing modalities.

## 5. Conclusions

In this paper, we propose ADAPT, a modality-agnostic representation framework designed to operate effectively under missing modalities during both training and testing. Our framework has been challenged on two different tasks targeting the detection of physiological changes, outperforming the current state of the art while showcasing its superiority for handling missing modalities. Extensive ablations indicate the robustness of our method on different scenarios and strategies. Future work includes applications to other medical tasks.

## Acknowledgments

This work was partially supported by Inria Action Exploratoire PreMediT (Precision Medicine using Topology) and the ANR-22-CE39-0016 APATE. Additionally, it was partly performed using HPC resources from GENCI-IDRIS (Grant 2023-AD011014747).

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

## Appendix A. `LOC` dataset

**$g$-forces and fighter pilots.** Earth's gravity, commonly referred to as $1g$, induces a constant acceleration. Fighter pilots, however, encounter much higher accelerations, reaching up to $9g$ along the z-axis. This force induces a blood shift towards the lower body, reducing blood perfusion in the brain. Without mitigation, pilots can experience altered consciousness, encompassing diminished peripheral vision, central vision loss, Almost Loss Of Consciousness (ALOC), and G-Induced Loss of Consciousness (GLOC) (Morrissette and McGowan, 2000), sometimes with fatal outcomes. Due to the eye's heightened sensitivity to hypoxia, initial symptoms are often visual. As retinal blood pressure drops below Intraocular pressure (usually 10–21 mm Hg), blood flow diminishes, impacting the retina initially in areas farthest from the optic disc and central retinal artery before progressing toward central vision. Fighter pilots train in centrifuges[1] to apprehend the effects of $+g_z$ accelerations and perform the Anti-$g$ Straining Maneuvers (AGSM). AGSM consist of muscle contractions in the lower limbs, synchronized with breathing exercices. These maneuvers aim to elevate arterial blood pressure to sustain blood volume in the brain (Pollock et al., 2021). Each training is monitored by doctors through physiological data acquired in real time: electrocardiograms (ECG), electromyograms of the quadriceps (EMG), the acoustic breathing (AUDIO), the pressure on the pedals (PAL), the visual field (VF)[2], and through video monitoring. Several factors (Pollock et al., 2021; Park et al., 2015; Nunneley and Stribley, 1979), may influence pilots' tolerance to $g$-forces, thus posing challenges for doctors in detecting alterations of consciousness, both in centrifuge simulations and in-flight. Moreover, expanding detection to real scenarios is not trivial. The modalities differ: the video and visual field data are impractical due to pilots' equipment (helmets and full-face $O_2$ masks) and technical constraints, respectively.

**Case study with medical doctors.** Given the rare availability of videos, we conducted a case study with a sample of doctors in charge of the aeromedical training. The study aimed to assess the efficacy of relying solely on biomedical signals for identifying loss of consciousness.

*Protocol.* Each doctor was presented with 20 complete launches accompanied by biomedical signals and tasked with labeling them as "Loss of consciousness" (Example in Figure 3) or "No loss of consciousness" (Example in Figure 4).

*Results.* Surprisingly, doctors accurately classified 55% of launches with a variance of 6.53% using only biomedical signals. However, when provided with videos, doctors achieved 100% accuracy in launch classification. This underscores the crucial role of video data and emphasizes the necessity, from a medical standpoint, to incorporate its robust representation.

**Positive and negative samples.** To detect *alteration of consciousness*, we train our models for binary classification (distinguishing between *altered* and *unaltered*) on $n$-seconds windows. Such a window $w_t=[t-n, t]$ is considered positive if there is some consciousness alteration period $(t_s, t_e)$ such that $t-n \geq t_s$ and $t \leq t_e$. Otherwise, $w_t$ is considered negative. We fix $n = 3.1sec$.

---

1. Equipment capable of reproducing the intensity and jolt of the $g$-forces accelerations experienced in high-performance fighter jet.

2. Self-filled in by pilots

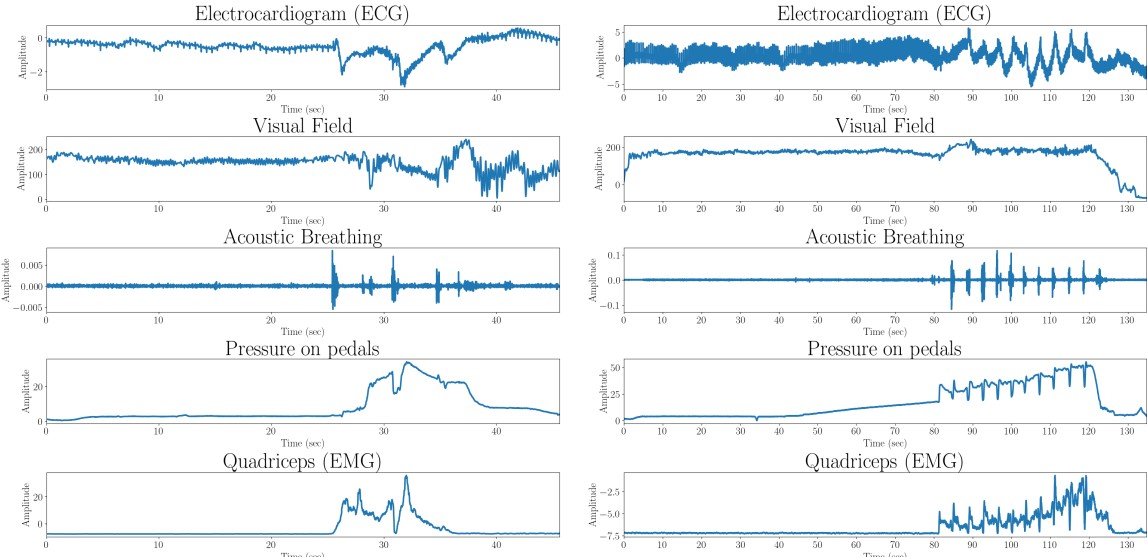

Figure 3: Example of a *Loss of consciousness* launch.

Figure 4: Example of a *No Loss of consciousness* launch.

## Appendix B. Implementations details.

### B.1. Architectural and training details.

|  |  | LOC | StressID |
|---|---|---|---|
| **Unimodal encoders** | Audio | BYOL-A (Niizumi et al., 2021) | BYOL-A (Niizumi et al., 2021) |
|  | Video | Hiera (Ryali et al., 2023) | Hiera (Ryali et al., 2023) |
|  | Biomedical Signals | InceptionTime (Ismail Fawaz et al., 2020) | FCN (Ismail Fawaz et al., 2019) |
| **Anchoring** | Anchor | Video | Video |
|  | $d$ | 128 | 64 |
|  | $\tau$ | Temperature schedule (Kukleva et al., 2023) | Temperature schedule (Kukleva et al., 2023) |
|  | $\gamma$ | 0.08 | 0.08 |
|  | projection head | 2 fully-connected layers | 1 fully-connected layer |
| **Multimodal Transformer** | $d_{model}$ | 64 | 64 |
|  | $n_{heads}$ | 4 | 4 |
|  | $N_L$ | 2 | 1 |
|  | $d_{ffn}$ | 256 | 256 |
|  | $d_v$ | 64 | 64 |
|  | $d_k$ | 64 | 64 |
|  | activation function | GeLU | GeLU |
|  | $\sigma$ | 0.02 | 0.1 |
|  | Normalization | LayerNorm | LayerNorm |
| **Linear Classifier** | loss | Balanced Cross Entropy | Cross Entropy |

Table 5: **Architectural details of ADAPT.**

Table 5 lists the architectural details used for each dataset. The seed is fixed to 1999. In our experiments, for both stages we use the AdamW (Loshchilov and Hutter, 2019) optimizer, with a weight decay of 0.05, a starting learning rate of $1e^{-4}$ following a cosine schedule, and preceded by a linear warm-up of 4 epochs on 4 NVIDIA Tesla V100 GPU using Pytorch (Paszke et al., 2019). The gradients' norms are clipped to 1, to ensure stability during training.

*Augmentations used for multi-view contrastive learning.* We used data augmentation with the sequential application, with each a 0.5 probability of Gaussian noise and modality dropout.

## B.2. Comparison to the state of the art.

We compare our approach directly with the methods utilized in Chaptoukaev et al. (2023), which established the state-of-the-art for `StressID`:

1. 'feature-level fusion': unimodal features features are combined into a single high-dimensional feature vector, used as input to a MLP trained with Cross-Entropy loss for `StressID` and Cost-Sensitive Cross Entropy (Huang et al., 2016) for `LOC` to tackle class imbalance and fair comparison with ADAPT.

2. 'decision-level fusion': independent SVMs are trained for each modality using the unimodal features as input, and integrate the results of the individual classifiers at the decision level, i.e. the results are combined into a single decision using ensemble rules. Four decision rules are proposed in Chaptoukaev et al. (2023): sum rule fusion, average rule fusion, product rule fusion and maximum rule fusion. The best out of the four decision rules is reported.

Additionally, we implemented ShaSpec (Wang et al., 2023a). ShaSpec maximizes the utilization of all available input modalities during training and evaluation by learning shared and specific features for better data representation. However, due to the varying dimensions of our input modalities (e.g., 3D video, 1D biomedical signals), employing a shared encoder is nontrivial. Hence, we adopt an adapted version, *ShaSpec+*, where encoded inputs are fed to the shared encoder instead of raw inputs. To ensure fair comparison with ADAPT, we use identical settings, including the same encoders: Hiera (Ryali et al., 2023) for video, Byol-a (Niizumi et al., 2021) for audio, and 1D CNN (Wang et al., 2023c) for biomedical signals. Additionally, to address class imbalance for the `LOC` dataset we substitute the cross-entropy loss with cost-sensitive cross-entropy (Huang et al., 2016).

## Appendix C. Complementary results

### C.1. Unimodal performances

Performances of unimodal encoders are provided in Table 6 for `LOC` and Table 7 for `StressID`.

### C.2. ADAPT's robustness to missing modalities

**Evaluation of ADAPT on $X_{\text{test}}^*$.** Table 8 assesses ADAPT across three modality scenarios, evaluating its robustness by removing one or two modalities from $X_{\text{test}}^*$ (i.e., samples where all modalities are available) and comparing the results to the baseline ($X_{\text{test}}^*$ without any modality removed). We calculate the differences ($\Delta$) for comparison. Overall, $|\Delta| < 3.2$, further highlighting ADAPT's robustness with full modality availability. Importantly, even though video-only performs better individually by a large margin, ADAPT maintains robust results when it is removed (row 3).

| | LOC | | | | | | | |
| | $X^*_{\text{test}}$ | | | | $X_{\text{test}}$ | | | |
| | **ACC** | **F1** | **TNR** | **TPR** | **ACC** | **F1** | **TNR** | **TPR** |
|---|---|---|---|---|---|---|---|---|
| Video | 87.1(1.2) | 98.0 (0.2) | 98.0(0.4) | 75.6(2.5) | - | - | - | - |
| Audio | 57.6(1.5) | 96.0(0.2) | 95.5(4) | 19.7(2.9) | 55.6 (1.8) | 83.2(3.2) | 71.6(5.1) | 39.6(6.3) |
| Visual Field | 66.2(3.4) | 92.8(2.5) | 89.5(4.4) | 43.0(9.4) | 72.9(4.0) | 92.1(3.0) | 85.8(5.2) | 60.1(7.2) |
| Pedals | 69.2(2.5) | 96.6(0.2) | 95.8(0.5) | 42.6(5.4) | 68.5(1.5) | 89.5(0.3) | 81.2(1.0) | 55.7(3.4) |
| Electrocardiograms | 60.3(3.4) | 96.1(0.8) | 95.0(1.5) | 25.5(6.8) | 55.9(3.0) | 93.9(1.7) | 88.9(3.0) | 23.0(7.8) |
| Electromyograms | 58.5(12.9) | 26.8(11.7) | 22.8(10.2) | 94.2(9.5) | 54.9(4.2) | 27.5(10.8) | 21.6(30.6) | 88.1(22.5) |

Table 6: **Performance of unimodal encoders for LOC.** Audio, VF, PAL, ECG, EMG performances are evaluated after the *anchoring* (i.e after the alignment to the anchor, the video). We report the results on $X_{\text{test}}$ and $X^*_{\text{test}}$ in form: mean(std).

| | StressID | | | |
| | $X^*_{\text{test}}$ | | $X_{\text{test}}$ | |
| | **ACC** | **F1** | **ACC** | **F1** |
|---|---|---|---|---|
| EDA | 58.0 (2.8) | 65.4 (4) | 64.0(2.2) | 64.1(2.1) |
| RR | 57.1(4.1) | 58.0(4.8) | 58.4 (3.0) | 58.0(3.4) |
| ECG | 55.6(3.6) | 39.8(7.3) | 55.5(2.2) | 48.7(4.0) |
| Audio | 59.9(6.2) | 66.9(9.1) | - | - |

Table 7: **Performance of unimodal encoders for StressID.** Audio, EDA, ECG and RR performances are evaluated after the *anchoring* (i.e after the alignment to the anchor, the video). We report the results on $X_{\text{test}}$ and $X^*_{\text{test}}$ in form: mean(std).

| | LOC | | | | | |
| | **ACC** | $+\Delta$ | **TNR** | $+\Delta$ | **TPR** | $+\Delta$ |
|---|---|---|---|---|---|---|
| | 67.4(1.3) | | 65.3(1.6) | | 69.5 (1.5) | |
| *real-life* | 61.9(7.2) | 5.5 | 70.1(2.1) | -4.8 | 61.4(10.4) | 8.1 |
| *no video* | 64.9(8.3) | 2.5 | 63.0(15.2) | 2.3 | 66.8(6.5) | 2.7 |
| *no audio* | 64.9(8.3) | 2.5 | 63.0(15.2) | 2.3 | 66.8(6.4) | 2.7 |

Table 8: **Evaluation of ADAPT on three modality scenarios on $X^*_{\text{test}}$ for LOC.** For each scenario, we remove one or two modalities from the test samples. We report mean and standard deviation for ACC, TNR, and TPR, and calculate the differences ($\Delta$) compared to tests without removed modality.

