# OpenReview forum: "ADAPT: Multimodal Learning for Detecting Physiological Changes under Missing Modalities"
_MIDL.io/2024/Conference — MIDL 2024 Poster_

### Official Review · Reviewer_6ssF · 2024-02-26

**Confidence:** 4
**Preliminary Rating:** 3
**Final Rating:** 4

**Summary:**

This paper addresses a key question in multimodal fusion for medical data, balancing modality contributions when limited data with potential missingness in the modalities is present in the dataset. They propose a new multimodal deep learning framework named AnchoreD multimodAl Physiological Transformer (ADAPT), which in its contrastive training aligns all modalities in the space of the "strongest"anchor  to learn a joint embedding space via contrastive training and a masked multimodal trasnformer to align modalities and address missing modalities in the masking scheme.

They evaluate on two different datasets, one for detection of stress in individuals induced by specific triggers and the next for fighter pilots’ loss of consciousness induced by g-forces. They evaluate the performance of the ADAPT model against several baselines to demonstrate improvements in classification performance.

**Strengths:**

The paper is well written, the experimental and methodological descriptions are clear, concise and easy to follow. Two separate datasets and tasks are used for evaluation, on which the proposed ADAPT method demonstrates consistent improvements. Although the architectures used are not novel per se, the utility of anchoring and masked multimodal training for addressing missing and unreliable modality data is an interesting contribution of the work.

**Weaknesses:**

1. The baselines being compared against seemed rather simplistic as described (In fact, very little description is available by way of architectural details). These are broadly classified as decision level (late fusion) and feature fusion (early/intermediate fusion?). On the other hand, here have been many proposed methods used in literature to achieve multimodal fusion and modality alignment, eg. using graph neural networks, CNNs, etc as deep learning methods or the classical models in (Chaptoukaev
et al., 2023), unsupervised learning (eg canonical correlation analysis).  It is unclear whether the proposed method would outperform these approaches for these tasks and datasets.

2. For the LOC dataset, it is really difficult to gauge whether the ADAPT method or any of the baselines is a clear winner based on all the metric comparison provided against individual modality predictors, especially the video modality. While I understand that the modality is rarely available in real-life, is it not used as the anchor (most important modality) for training ADAPT and is assumed to be present during training?

3. For the missing modalities experiment, if my understanding is correct, the modalities are considered missing only at test time and not during training (Tables 2, 7, 8). For a real evaluation of the impact of missing modalities, I would expect a comparison where the percentage of the modality data is considered missing both at train and test time and varied systematically, which is the more realistic setting to evaluate the robustness and success of ADAPT at addressing this issue.

**Detailed Comments:**

Additional Questions:

1. What are the class distributions for both the datasets? Is there any class imbalance, and if so, how is it addressed in this work? Does this contribute to the difficulty in training the DecisionFusion and FeatureFusion baselines?
2. The authors should consider reporting the area under the precision recall curve (AU-PRC) to allow for easier comparison across methods, even when the dataset may potentially be unbalanced.
3. The presentation and font of Table 4 makes it really hard to parse.

**Justification Of Final Rating:**

I thank the authors for the detailed clarifications, additional experimental results and explanation of dataset characteristics for this application. Given that they have addressed my concerns adequately and improved the quality of the submission, I am willing to raise my score to 4

**Justification Of The Preliminary Rating:**

While the methodological contribution of the paper is interesting, it is unclear from the current experiments whether the proposed approach is indeed a good solution for multimodal fusion under the missing modality scenario. Therefore, my current rating for this paper is as a borderline submission.

**Questions To Address In The Rebuttal:**

Please refer to the points in weaknesses, especially the evaluation of robustness to missing modalities.

---

> ### Author Response · Authors · 2024-03-18
>
> We thank the reviewer for the insightful comments and questions. The major concerns regarding the challenges posed by the LOC dataset and how ADAPT addresses them were addressed in the comments to the submission.
>
> **W1. "The baselines compared against seemed rather simplistic[...]."**  This point is addressed in R2's comment.
>
> **W2. "For the LOC dataset, it is really difficult to gauge whether the ADAPT [...]."** This point is addressed in the main comment.
>
> **W3. “[...] For the missing modalities experiment, if my understanding is correct, the modalities are considered missing only at test time and not during training (Tables 2, 7, 8). [...]."** Relates also to **R1.W4 "The experiment lacks the effect of different ratios of modality incompleteness[...]."**
> ADAPT and robustness to missing modalities. StressID and LOC datasets exhibit missing training and test sets modalities. StressID presents 18% of observations with missing video and 46% with audio missing, while LOC suffers from a significant 90% absence of videos across its dataset. We introduce a dedicated paragraph in Section 4, Subsection 4.1, to clarify the missing modalities across datasets. In this work, we focus on examining ADAPT's robustness to various subsets of modalities during inference (Table 2) while training with the inherent missing modalities present in both datasets. This approach aims to underscore ADAPT's real-life applicability. Our results demonstrate that ADAPT maintains competitive performance across the three studied modality scenarios: no video, no audio, and no audio with no video.
>
> Next, we answer point by point the questions raised by the reviewer:
>
> **Q1. "What are the class distributions for both the datasets? [...]"**
> We apologize if this aspect wasn't clear in our initial submission. The LOC dataset exhibits a class imbalance of {1:50}, while the StressID dataset does not suffer from any imbalance. The imbalance ratio for the LOC dataset was specified in the metrics paragraph of subsection 4.1. In this study, we mitigate class imbalance using cost-sensitive learning, specifically employing the Cost-Sensitive Cross Entropy loss weighted by inverse class frequency (Huang et al., 2016; Wang et al., 2017). This loss function is utilized to train the classifiers of ADAPT, unimodal baseline, ShaSpec+, and Feature Fusion. In the revised version of the manuscript (Section 4, Subsection 4.1), we provide enhanced clarity by highlighting the imbalance ratio in the dataset presentation and adding further details on our approach to address it.
> ADAPT effectively handles class imbalance, achieving the highest True Positive Rate (TPR) on LOC among multimodal methods. Moreover, it tends to strike a superior balance between True Negative Rate (TNR) and TPR, as illustrated in Figure 2 and emphasized in the opening paragraph of section 4, subsection 4.2.
>
> **Q2. "The authors should consider reporting the area under the precision recall curve (AU-PRC) to allow for easier comparison across methods."**
> In this paper, we assess ADAPT using Class Balanced Accuracy, True Positive Rate (TPR), and True Negative Rate (TNR). These metrics are highly robust for evaluating performance on imbalanced datasets (Huang et al., 2016; Luque et al., 2019). While we initially considered precision, it has been shown to exhibit significant bias in the context of highly imbalanced datasets (Luque et al., 2019). Consequently, the AU-PRC would similarly suffer from this bias. Specifically, due to the prevalence of true negatives, the fraction of false positives may significantly outweigh the number of true positives, leading to a large denominator in the precision calculation and consequently deflating the precision due to false negatives.
>
> **Q3. "The presentation and font of Table 4 makes it really hard to parse."**
> We appreciate the reviewer's feedback. The revised manuscript now includes an updated table (Section 4, Subsection 4.2).

---

> > ### Comment · Reviewer_wJd1 · 2024-03-18
> >
> > Thanks for addressing those concerns. I would accept it now.

---

### Official Review · Reviewer_wJd1 · 2024-02-26

**Confidence:** 4
**Preliminary Rating:** 5
**Recommendation:** Oral
**Final Rating:** 5

**Summary:**

Focusing on detecting alterations of physiological responses in two real-life scenarios, namely: stress in individuals induced by specific triggers and fighter pilots’ loss of consciousness induced by g-forces.

**Strengths:**

This concept is unique and holds great potential for medical applications. The structure and language of the paper are fitting, enhancing its clarity and impact. It's a commendable work that can significantly contribute to healthcare advancements.

**Weaknesses:**

While the paper is short and concise, there are sections with grammatical errors that need addressing. Additionally, it lacks citations from recent papers, which is crucial for grounding the research in the current state of the field. These improvements are necessary for a more robust submission.

**Detailed Comments:**

Nothing

**Justification Of Final Rating:**

The final rating reflects a comprehensive evaluation of the author's response to feedback throughout the revision process. Each comment was meticulously considered, leading to significant improvements in the clarity, depth, and accuracy of the work. The author's dedication to refining their manuscript based on the constructive critiques provided is commendable. This thorough engagement not only enhanced the overall quality of the submission but also demonstrated the author's commitment to excellence and receptiveness to feedback. The enhancements made as a result of addressing the comments have substantially contributed to the work's coherence, making it a valuable contribution to its field. Therefore, the final rating is justified by the author's successful efforts to integrate all provided feedback, ultimately elevating the work's academic and practical implications.

**Justification Of The Preliminary Rating:**

I would accept this paper. The content is intriguing and presents a fresh perspective that contributes significantly to its field. The methodology is sound, and the results are compelling. Overall, a valuable addition to the academic discourse

**Questions To Address In The Rebuttal:**

Nothing

**Special Issue:**

Yes

---

> ### Author Response · Authors · 2024-03-18
>
> We thank the reviewer for their positive evaluation of our work and address point by point the weaknesses raised by the reviewer:
>
> **W1. “It lacks citations from recent papers.”**
> Relates also to **R1.W2. “Experiments lack a detailed comparison with other state-of-the-art[...]”** and to **R3.W1. “The baselines compared against seemed rather simplistic[...]”**
>
> We compare our approach directly with the methods utilized in (Chaptoukaev et al., 2023), which established the state-of-the-art for the StressID dataset. While acknowledging the importance of further comparisons, current methodologies exhibit limitations due to inherent dataset challenges. As detailed in the revised Section 2 of the manuscript, these methods suffer from (1) limited generalizability to more than two modalities (Ma et al., 2021), (2) bias towards the most abundant modalities leading to suboptimal performance, as observed in the severely underrepresented video modality in the LOC dataset (Konwer et al., 2023), and (3) an assumption of homogeneity among modalities, complicating extension to heterogeneous modalities like imaging and biomedical signals by utilizing shared encoders with uniform input dimensions (Wang et al., 2023).
> Following the reviewer's suggestions, we compare against an additional state-of-the-art method, ShaSpec (Wang et al., 2023), a SOTA method that outperforms SMIL (Ma et al., 2021) on Audiovision-MNIST, altered to be compatible with our datasets. ShaSpec maximizes the utilization of all available input modalities during training and evaluation by learning shared and specific features for better data representation. However, due to the varying dimensions of our input modalities (e.g., 3D video, 1D biomedical signals), employing a shared encoder is nontrivial. Hence, we adopt an adapted version, ShaSpec+, where encoded inputs are fed to the shared encoder instead of raw inputs. To ensure a fair comparison with ADAPT, we use identical settings, including the same encoders (Hiera (Ryal et al. 2023) for video, Byol-a for audio, and 1D CNN for biomedical signals). Additionally, to address class imbalance for the LOC dataset, we substitute the cross-entropy loss with cost-sensitive cross-entropy (Huang et al. 2016).
> The Table (Table 1, Section 4 in the manuscript) below demonstrates the results. In the case of complete modality during testing, ADAPT demonstrates superior performance compared to both methods (Chaptoukaev et al., 2023) and the additional comparison with ShaSpec+ (Wang et al., 2023). Although ShaSpec+ yields competitive results on the Stressed dataset, it achieves the lowest true positive rate (TPR) (7.3%) on the LOC dataset, highlighting the relevance of ADAPT in extreme scenarios. Our experiments reveal ADAPT's potential to handle missing modalities robustly, even in extreme imbalance (e.g., 1:50 class ratio for LOC) or severe modalities missing during training.
>
> **W2. “Sections with grammatical errors”**
> Thank you for noticing this. Authors proofread several times the manuscript and corrected grammatical errors in the updated pdf.

---

> > ### Comment · Reviewer_wJd1 · 2024-03-18
> >
> > Thanks for addressing those concerns. I would accept it now.

---

### Official Review · Reviewer_UHJs · 2024-03-07

**Confidence:** 3
**Preliminary Rating:** 2
**Final Rating:** 2.5

**Summary:**

The proposed framework is to develop a modality-agnostic representation for multimodal physiological data by training modality-specific encoders with a contrastive learning objective to align their representations to an anchor modality. This is designed
to operate effectively under missing modalities during both training and testing.

Experiments conducted in the study demonstrate the effectiveness of ADAPT in maintaining high-quality results even when certain modalities are missing, showcasing its robustness and capability to align representations in the absence of specific modalities. The strength of ADAPT lies in its ability to address the challenges of missing modalities in multimodal learning, providing a solution for real-life applications where data collection may be incomplete.

**Strengths:**

1. The paper focuses on the challenge of missing modality, and allows to operate different experiment settings, *i.e.*, missing modalities during both training and testing.
2. This framework aligns multimodal representation to a common feature space for better fusion and further missing modality task.
3. Extensive experiments on ADAPT demonstrate the effectiveness of the proposed framework in handling missing modalities and outperforming existing methods.

**Weaknesses:**

1. I think the challenge of missing modality is not only a task to handle incomplete modality settings in both train and test stages, but also to try to reduce the effect of missing modality as much as we can. This paper focuses on the former one and just masks the missing modality's representation feature, which is not reasonable.
2. Experiments lack a detailed comparison with other state-of-the-art missing modality frameworks. Only two comparisons in one paper are insufficient.
3. The choice of anchor is unclear, can we use other modalities and how do they perform? Like, for the LOC dataset, most of the video data is missing; why not choose another modality to replace it? Please explain it from a theoretical and experimental view.
4. The experiment lacks the effect of different ratios of modality incompleteness on the experimental results. The relevant experiment setting can refer to *SMIL: Multimodal Learning with Severely Missing Modality (AAAI 2021)*.

**Detailed Comments:**

Please see strengths and weakness.

**Justification Of Final Rating:**

Thanks for addressing my question. Authors provide clear comments, additional experiments. Though I think the idea of anchor is interesting, why choose video as an anchor is still unclear. I am willing to raise my score to 3r.

**Justification Of The Preliminary Rating:**

The "Weak reject" rating is justified based on the submission's shortcomings. The content lacks depth, offering limited insights into the missing modality.  And it lacks ablation studies in experiments, with critical elements either incomplete or unclear. To improve the ratings, please address the weaknesses and questions.

**Questions To Address In The Rebuttal:**

1. In common sense, the multimodal data has more information than unimodal data, which is helpful in real-life decisions. However, unimodal dataset outperformed the multimodal settings in Tab. 1.
2. Many modality-specific encoders are applied in the proposed frameworks. How can they influence the performance of this network?
3. Since the proposed method uses a masked Transformer, how does the maskx c operation influence the performance?

For other questions, please see weakness.

---

> ### Author Response · Authors · 2024-03-18
>
> The primary concern relating to the role of the video and, more generally, the anchor in our overall architecture has been addressed in our main comments to the reviewers. Substantial revisions were applied to the manuscript in the introduction, the related work (Section 2), and the section relating to the datasets (Section 4, Subsection 4.1).
> We answer the different questions and weaknesses raised by the reviewer point by point.
>
> **W1. “I think the challenge of missing modality is not only a task to handle incomplete train and test stages, but also to try to reduce the effect of missing modality as much as we can. [...].”**
> We thank the reviewer for raising this important point regarding the challenges inherent to missing modalities in multimodal learning. Our method effectively manages and mitigates the impact of missing modalities, as demonstrated in Table 1 (last row) and Table 2 (first row). ADAPT consistently performs well across datasets, regardless of full modality presence or missing modalities.
>
> || LOC || StressID ||
> |:-| :-|:-|:-|:-|
> | | **ACC** | **F1** | **ACC** | **F1** |
> |full modality | 67.4(1.2) |76.9(2.5)| 69.5(3.7) | 75.9(4.3)|
> | with missing modalities | 66.2(3.3)| 77.7(4.3) | 69.5(2.9)|70.0(3.1)|
>
> Moreover, ADAPT maintains stability even when a modality is entirely absent from the test set, as shown in Table 2, underscoring its resilience.
> Architecturally, ADAPT achieves robustness through two key mechanisms detailed in Section 2: anchoring and modality dropout. Anchoring involves embedding all modalities into the same latent space, mitigating biases towards the most available modalities, and ensuring scalability. We select video as the anchor due to its strong unimodal performance, even with significant missing data (e.g., video exhibits a missing ratio of 90% for LOC and 18% for StressID), leveraging its strong priors for downstream tasks where video may be unavailable. Additionally, we employ multi-view contrastive learning and a tailored data augmentation technique for multimodal learning: Modality dropout (Shi et al., 2022). Modality dropout mitigates the model’s reliance on any single modality, enhancing its robustness by randomly considering missing features of some modalities.
> All in all, ADAPT effectively addresses the handling of missing modalities while ensuring robustness.
>
> **W2. “Experiments lack a detailed comparison with other state-of-the-art[...].”** This point is addressed in the comment to R2.
>
> **W3. “The choice of anchor is unclear [...]”** This point is addressed in the general comment.
>
> **W4. “The experiment lacks the effect of different ratios of modality incompleteness[...].”** This point is addressed in the comment to R3.
>
> **Q1. “In common sense, the multimodal data has more information than unimodal data [...].”** This point is addressed in the general comment.
>
> **Q2. “Many modality-specific encoders are applied in the proposed frameworks.”**
> Our framework relies on a separate stream architecture rather than a single stream, motivated by several factors: (1) it enables using foundational models for aligning all modalities to the anchor (e.g., video); (2) using a shared encoder is challenging due to the heterogeneous modalities considered for both datasets and more precisely varying dimensions in modality inputs, such as 3D video and 1D biomedical signal. We have clarified this choice in the related work section of the revised manuscript to provide better insight into our approach. Regarding the selection of the encoders, we utilized state-of-the-art methods for each modality: Hiera (Ryal et al., 2023) for video, BYOL-A for audio (Niizumi et al., 2021), and 1D CNN for biomedical signals (Wang et al., 2023a). Additionally, for comparison, we considered using the handcrafted features proposed in (Chaptoukaev et al., 2023) in Table 4; however, as highlighted in row 3, it leads to suboptimal performances compared to ADAPT (row 6). Furthermore, we also employed our anchored unimodal encoders (rows 4 & 6) with (Chaptoukaev et al., 2023) fusion methods. It outperforms the fusion methods with handcrafted features by a large margin (+ 5% ACC and +6% in F1).
> In summary, our experiments highlight the powerful representations of our anchored unimodal encoders and, more generally, our separate stream architecture.
>
> **Q3. “Since the proposed method uses a masked Transformer [...]”**
> In this work, we choose to leverage the mask in the attention mechanism from the transformer architecture. This choice is inspired by prior successful implementations (Milecki et al., 2022; Ma et al., 2022) in tasks such as follow-up patient examination completion and handling missing modalities in bimodal datasets.

---

> > ### Comment · Reviewer_wJd1 · 2024-03-18
> >
> > Thanks for addressing those concerns. I would accept it now.

---

> > ### Comment · Reviewer_UHJs · 2024-03-22
> > **Official Comment by Reviewer UHJs**
> >
> > Thanks for addressing those concerns, I would change the rating.

---

> > > ### Comment · Area_Chair_TeSb · 2024-03-30
> > >
> > > Dear Reviewer,
> > >
> > > Could you update your final rating for this paper?
> > >
> > > Best

---

### Author Response · Authors · 2024-03-18

We would like to thank the reviewers for evaluating our work and for their insightful comments and queries. We provide here additional clarifications and address the main comments. A detailed response to each reviewer is also given (R1 - UHJs, R2 - wJd1, R3 - 6ssF).
We have submitted a revised version of the manuscript, which addresses the points highlighted by the reviewers and completes the bibliography. Modifications made to the manuscript are written in blue.


**R1 and R3 expressed concerns about the challenges posed by the LOC dataset and how ADAPT addresses them**, both in terms of multimodal architecture and the anchor choice. We addressed this issue more explicitly in our answer and the revised manuscript.

(i) *The extreme case of the LOC dataset.* The LOC dataset presents several extreme characteristics: (1) a severe class imbalance (1:50), (2) a high proportion of missing videos (present in only 10% of observations), while demonstrating strong unimodal performances compared to biomedical signals, (3) and impracticality in real-life data collection scenarios.
Given the rare availability of videos, we conducted a case study with a sample of doctors in charge of the aeromedical training. The study aimed to assess the efficacy of relying solely on biomedical signals for identifying loss of consciousness. Each doctor was presented with 20 complete launches accompanied by biomedical signals and tasked with labeling them as "Loss of consciousness" or "No loss of consciousness." Surprisingly, doctors accurately classified **only** 55% of launches with a variance of 6.53% using only biomedical signals. However, when provided with videos, doctors achieved 100% accuracy in launch classification (details on the case study were added in Appendix A). This underscores the crucial role of videos and emphasizes the necessity, from a medical standpoint, to incorporate their robust representation. Therefore, leveraging the strong priors of video data during training is imperative for achieving robust results on the LOC dataset, even in scenarios where video data is utterly absent during inference.
As demonstrated in Table 1, ADAPT effectively leverages video priors, surpassing other multimodal approaches and maintaining strong performance even when video data is absent from the test set (Table 2). Additionally, on more balanced datasets like the StressID dataset (with ½ positive and negative ratio), ADAPT consistently exhibits strong multimodal performance, outperforming unimodal methods and the previous state-of-the-art established by Chaptoukaev et al. (2023).

(ii) *The choice of anchor.* Anchoring facilitates the integration of all modalities into a unified feature space, enhancing training efficacy and ensuring linear scalability for joint representation. By aligning all modalities to a single representation (our anchor), we mitigate the quadratic complexity inherent in pairwise consideration of all modalities. As Konwer et al. (2023) highlighted, the absence of specific modalities in some observations can bias the shared latent space toward more available ones. Despite video being absent in 90% of samples in the LOC dataset, it exhibits robust unimodal performance when employing a pre-trained encoder. This observation motivates choosing video as the anchor to leverage its strong representation in the shared latent space. Table 3 illustrates the pivotal role of video. This choice yields a noteworthy increase of +12% in ACC on the LOC dataset while maintaining F1 scores and a +4% improvement in both ACC and F1 on the StressID dataset compared to ADAPT without anchored unimodal encoders (row 1). Moreover, as discussed in Section 3, any modality can serve as the anchor. To further support this assertion, we conduct additional experiments using audio as the anchor, for which we can also leverage a pre-trained encoder, yet consistently observe more robust results with video as the anchor (we modify Table 3 accordingly in the manuscript).
In summary, the choice of the anchor is driven by its robust unimodal performance, which remains effective even in the presence of high missingness.

---

> ### Comment · Reviewer_wJd1 · 2024-03-18
>
> Thanks for addressing those concerns. I would accept it now.

---

### Meta-Review · Area_Chair_TeSb · 2024-04-03

**Recommendation:** Accept (Poster)
**Confidence:** 5

**Metareview:**

This paper received the final ratings with 1 borderline reject, 1 strong accept and 1 weak accept. The authors addressed most of the comments during the rebuttal while some justification and details remain to be further illustrated. After reconciling the comments, AC tend to accept the paper and strongly suggest the authors take the reviewers’ comments into the final revision.

---

### Decision · Program_Chairs · 2024-04-05

Accept (Poster)